# DTG-SSOD: Dense Teacher Guidance for Semi-Supervised Object Detection

**Gang Li**[1,2], **Xiang Li**[3]*, **Yujie Wang**[2], **Yichao Wu**[2], **Ding Liang**[2], **Shanshan Zhang**[1]*

[1]Nanjing University of Science and Technology  [2]SenseTime Research  [3]Nankai University

{gang.li, shanshan.zhang}@njust.edu.cn, xiang.li.implus@nankai.edu.cn
{wangyujie, wuyichao, liangding}@sensetime.com

## Abstract

The Mean-Teacher (MT) scheme is widely adopted in semi-supervised object detection (SSOD). In MT, the *sparse* pseudo labels, offered by the final predictions of the teacher (e.g., after Non Maximum Suppression (NMS) post-processing), are adopted for the *dense* supervision for the student via hand-crafted label assignment. However, the "sparse-to-dense" paradigm complicates the pipeline of SSOD, and simultaneously neglects the powerful direct, dense teacher supervision. In this paper, we attempt to directly leverage the dense guidance of teacher to supervise student training, i.e., the "dense-to-dense" paradigm. Specifically, we propose the Inverse NMS Clustering (INC) and Rank Matching (RM) to instantiate the dense supervision, without the widely used, conventional sparse pseudo labels. INC leads the student to group candidate boxes into clusters in NMS as the teacher does, which is implemented by learning grouping information revealed in NMS procedure of the teacher. After obtaining the same grouping scheme as the teacher via INC, the student further imitates the rank distribution of the teacher over clustered candidates through Rank Matching. With the proposed INC and RM, we integrate Dense Teacher Guidance into Semi-Supervised Object Detection (termed "DTG-SSOD"), successfully abandoning sparse pseudo labels and enabling more informative learning on unlabeled data. On COCO benchmark, our DTG-SSOD achieves state-of-the-art performance under various labeling ratios. For example, under 10% labeling ratio, DTG-SSOD improves the supervised baseline from 26.9 to 35.9 mAP, outperforming the previous best method Soft Teacher by 1.9 points. Code will be released at: https://github.com/ligang-cs/DTG-SSOD.

## 1 Introduction

Thanks to plenty of annotated data, deep learning based computer vision has achieved great success in the past few years [1–4]. However, labeling accurate annotations is usually labour-consuming and expensive, especially for dense prediction tasks [5–8], such as object detection, where multiple bounding boxes with their corresponding category labels need to be annotated extensively for each image. Some works attempt to leverage easily accessible unlabeled data to promote object detection in a semi-supervised manner [9–12]. The model is first initialized by learning on the labeled data. Then it is adopted for generating pseudo labels (i.e., pseudo bounding boxes and category labels) on the unlabeled data for training.

Recently, advanced Semi-Supervised Object Detection (SSOD) methods follow the Mean-Teacher paradigm [14], where a stronger teacher model is built from the student model via Exponential Moving

---

*Corresponding author.

36th Conference on Neural Information Processing Systems (NeurIPS 2022).

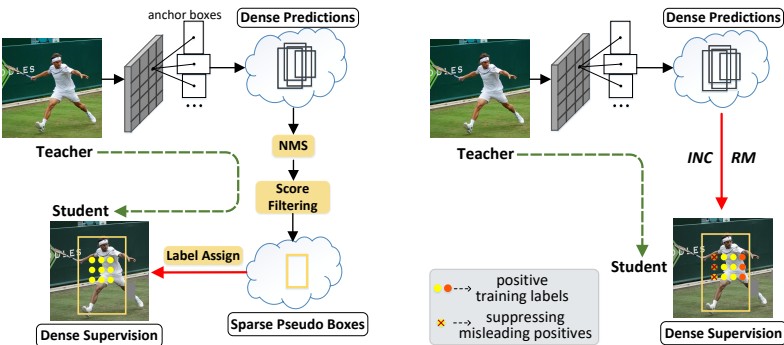

Figure 1: Comparison of two paradigms: (a) previous methods [9, 10, 13] perform NMS and score filtering on teacher's dense predictions to obtain *sparse* pseudo boxes, which are further converted to *dense* supervision of the student via the label assignment; (b) our DTG-SSOD directly converts teacher's *dense* predictions into the *dense* supervision for the student, by the proposed INC and RM.

Average (EMA) to produce supervision signals over unlabeled data. Nevertheless, all of the previous methods [9–11, 13, 12, 15] adopt *sparse* pseudo labels after the post-processing (e.g., NMS and score filtering) as the supervision, passed from the teacher to student. To guarantee the precision of pseudo labels, a high score threshold (e.g., 0.9) is usually selected to filter coarse detection boxes and only a few pseudo boxes could survive. Through the hand-crafted label assignment strategy [16–18], the detector transfers *sparse* pseudo labels into *dense* supervision for student training, which is termed the "sparse-to-dense" paradigm. Such sparse pseudo labels neglect some important properties of object detection (e.g., dense predictions, contextual relations between neighbouring samples), and lack of informative and dense teacher guidance, leading to limited benefits for learning on the unlabeled data. In Fig. 1, we illustrate the "sparse-to-dense" paradigm, where the sparse pseudo labels originated from the dense predictions of teacher are adopted for the dense supervision of student. This process involves many handcrafted components (e.g., NMS, score thresholding, and label assignment), which inevitably introduce accumulated noise/bias to supervisory signals for student. Therefore, it is more intuitive to adopt the "dense-to-dense" paradigm, i.e., directly employing *dense* predictions of teacher to guide the training of student, whilst abandoning the intermediate operations.

In this work, to implement the "dense-to-dense" paradigm, we propose to enforce the student to mimic the behavior of the teacher model. Specifically, the behaviour is motivated from the procedure how dense samples perform NMS, an essential post-processing scheme for popular detectors. To regularize the consistency on NMS behaviour between the teacher and student, we introduce Inverse NMS Clustering (INC) and Rank Matching (RM). INC leads the student to group the candidate boxes into clusters in NMS as the teacher does. In the process of NMS, candidates are grouped into multiples clusters. Within each cluster, teacher candidates are supposed to predict the identical object, which will assign training labels to the corresponding student samples in turn. Different from the previous sparse-to-dense paradigm, where sparse pseudo labels are converted to student training labels via the hand-crafted label assignment, our dense-to-dense paradigm assigns dense training labels based on grouping information by reversing the NMS process of the teacher. Based on INC where the student obtains the same grouping scheme of NMS with the teacher, we further introduce Rank Matching to align the score rank over the clustered candidates between teacher and student. NMS suppresses the duplicated detection and reserves only one box for each cluster through the rank of clustered candidates. Therefore, the rank of the samples within each cluster could contain rich relationship learned by the teacher and it can behave as informative dense supervision which guides the student to reserve as the same candidates as the teacher during NMS.

Through the proposed INC and RM, the teacher model provides not only fine-grained and appropriate training labels, but also informative rank knowledge, realizing Dense Teacher Guidance for SSOD (DTG-SSOD). To validate the effectiveness of our method, we conduct extensive experiments on COCO benchmark under Partially Labeled Data and Fully Labeled Data settings. Under the Partially Labeled Data, where only 1%,2%,5%, and 10% images are labeled, the proposed DTG-SSOD achieves the state-of-the-art performance, for example, under 5% and 10% labeling ratios, DTG-SSOD surpasses the previous best method Soft Teacher by +1.2 and +1.9 mAP, respectively. Under the Fully Labeled Data, where the unlabeled2017 set is used as unlabeled data, the proposed DTG-

SSOD not only achieves the best performance (45.7 mAP), but also improves the learning efficiency, e.g., DTG-SSOD halves the training iterations of Soft Teacher but achieves even better performance.

## 2 Related Work

**Semi-Supervised Image Classification (SSIC)** The pseudo labeling [19–21] is a popular pipeline, where unlabeled data is labeled by the model itself, and then combined with labeled data to act as training examples in the cycle self-training manner. In the pseudo-labeling, hard pseudo labels are usually used with a confidence-based thresholding where only images with high confidence can be retained. Recently, some improvements [22–24] on pseudo-labeling are proposed, for example, Curriculum Labeling [22] proposes that restarting model parameters before each training cycle can promote self-training; Noisy Student Training [23] adopts an equal-or-larger model with noise injected to learn the pseudo labels, which are generated in the last self-training cycle. Apart from hard pseudo labels, Noisy Student Training also tried soft pseudo labels, observing that for in-domain unlabeled images, both hard and soft pseudo labels perform well, while for out-of-domain unlabeled data, soft pseudo labels can work better. On the other hand, based on the insight that the model should be robust to input perturbations, UDA [25] designs advanced data augmentations and enforces the model to predict the similar class distributions for the differently augmented inputs. Here, soft labels (i.e., distributions) are superior in measuring the distance between different model predictions over the hard ones. As another strong method, FixMatch [26] introduces a separate weak and strong augmentation and converts model predictions into hard labels, since the labels with low entropy are believed to be beneficial in their pipeline. To conclude, previous SSIC works investigate various forms of supervisory signals (including hard and soft pseudo labels), and observe the form of supervision signals matters in semi-supervised learning, which inspires us to investigate the optimal supervision form for semi-supervised object detection.

**Semi-Supervised Object Detection (SSOD)** CSD [27] is an early attempt to construct consistency learning for horizontally flipped image pairs, where only the flip transformation can be used as the augmentation, which greatly limits its performance. STAC [28] borrows the separate weak and strong augmentation from FixMatch [26] and applies it to SSOD, obtaining promising performance. After that, [10, 11, 13, 9, 29] abandon the conventional multi-phase training and update the teacher model from the student via EMA after each training iteration. Apart from the framework, previous works promote SSOD from two main perspectives: designing advanced data augmentation and improving the precision of pseudo labels. For the former one, Instant-Teaching [10] introduces more complex augmentations for unlabeled data, including Mixup and Mosaic; MUM [15] designs a Mix/UnMix data augmentation, which enforces the student to reconstruct unmixed feature for the mixed input images. As for the latter, through ensembling the teacher model on input images and prediction head, respectively, Humble Teacher [11] and Instant-Teaching [10] reduce confirmation bias of the pseudo labels; Soft Teacher [9] selects reliable pseudo boxes by averaging regression results from multiple jittered boxes; Rethinking Pse [12] converts the box regression to a classification task, making localization quality easy to estimate. Different from these methods, which all adopt sparse pseudo boxes with category labels as the supervisory signal, our method investigates the new form of supervision for the first time and proposes to replace sparse pseudo labels with more informative dense supervision.

## 3 Method

The mechanism of weak-strong augmentation pairs shows promising performance in SSOD. Based on it, we first build an end-to-end pseudo-labeling baseline, described in Sec. 3.1, through which we explain the task formulation of SSOD and the conventional sparse-to-dense paradigm. Then in Sec. 3.2, we introduce the proposed Inverse NMS Clustering (INC) and Rank Matching (RM), i.e., the key two components of the proposed DTG-SSOD.

### 3.1 Sparse-to-dense Paradigm

**Task Formulation.** The framework of SSOD is illustrated in Fig. 2. The Mean-Teacher scheme is a common practice for previous works [9, 10, 13, 11] and realizes the end-to-end training, where the teacher is built from the student via EMA after each training iteration. The teacher takes weakly

augmented (e.g., flip and resize) images as input to generate pseudo labels, while the student is applied with strong augmentations (e.g., cutout, geometry transformation) for training. Strong and appropriate data augmentations play an important role, which not only increase the difficulty of student tasks and alleviate the over-confidence issue [25], but also enable the student to be invariant to various input perturbations, leading to robust representation learning [23, 26]. In this work, we borrow the data augmentations from Soft Teacher [9], and more details can be found in Appendix. Each training batch consists of both labeled data $\{x_i^l, y_i^l\}_{i=1}^{N_l}$ and unlabeled data $\{x_i^u\}_{i=1}^{N_u}$, which are mixed according to the ratio $r = \frac{N_l}{N_u}$, where $N_l$ and $N_u$ are the number of labeled and unlabeled images in each batch. The overall loss function of SSOD is formulated as:

$$\mathcal{L} = \mathcal{L}^l + \alpha \mathcal{L}^u, \tag{1}$$

where $\alpha$ controls the contribution of losses on labeled data $\mathcal{L}^l$ and unlabeled data $\mathcal{L}^u$.

**Sparse-to-dense Baseline.** All previous SSOD approaches [9, 10, 13, 11] are based on the sparse-to-dense mechanism, where sparse pseudo boxes with category labels are generated to behave as ground-truths for the student training. It comes with confidence-based thresholding where only pseudo labels with high confidence (e.g., larger than 0.9) are retained. This makes foreground supervision on unlabeled data much sparser than that on labeled data, therefore, the class imbalance problem is amplified in SSOD and impedes the detector training severely. To alleviate this issue, we borrow some strengths of previous works: Soft Teacher [9] sets the mixing ratio $r$ to 1/4 to sample more unlabeled data in each training batch, which makes the number of foreground samples on unlabeled data approach that on labeled data; Unbiased Teacher [13] replaces the cross-entropy loss with the Focal loss [16], reducing the gradient contribution from easy examples. Both of these two improvements, i.e., the appropriate mixing ratio $r$ (1/4) and Focal loss, are adopted in the sparse-to-dense baseline and our dense-to-dense DTG method. Because the teacher only provides *sparse* pseudo labels, which are further converted into the *dense* supervision for the student training, these methods are termed "sparse-to-dense" paradigm. In theory, our SSOD method is independent of the detection framework and can be applicable to both one-stage and two-stage detectors. For a fair comparison with previous works, we use the Faster RCNN [17] as the default detection framework.

## 3.2   Our method: Dense-to-dense paradigm

Pseudo labels in SSOD contain both bounding boxes and category labels. Considering that classification scores are not strongly correlated with localization quality [30, 3, 31, 32], although confidence-based thresholding can filter out many false positives, the localization quality of pseudo boxes is not guaranteed. Poorly-localized pseudo boxes will mislead the label assignment and confuse the decision boundary. Therefore, previous methods [12, 9] design two separate sets of indicators to measure the confidence of categories and box locations, which complicates the SSOD pipeline. Moreover, the sparse-to-dense paradigm neglects some important properties of object detection and lacks of powerful direct, dense teacher guidance. Motivated by these, we propose to directly employ teacher's dense predictions to supervise the student's training, termed "dense-to-dense" paradigm, abandoning the explicit sparse pseudo boxes.

Dense predictions widely exist in both RPN and RCNN stages. Specifically, in the RPN stage, multiple anchors are usually tiled on each feature map grid [17, 16]; while in the RCNN stage, taking 2,000 proposals as training samples is a common practice [17]. Compared against the sparse pseudo labels, dense predictions of the teacher contain rich knowledge, and we attempt to pass it to the student by enforcing the student to mimic the dense behaviour of the teacher. Specifically, we analyze how teacher's dense samples perform to eliminate redundancy in the process of Non Maximum Suppression (NMS). Based on the analysis, we design Inverse NMS Clustering and Rank Matching to regularize consistency on the NMS procedure between the teacher and student, from which the student can obtain precise training labels and informative supervision signals.

However, implementing dense mimic is not trivial. Due to the separate weak-strong augmentations, one-to-one correspondence between the samples of the teacher and student can not be easily identified. We design a simple yet effective matching strategy to address this. On the R-CNN stage, to keep the same proposals between the teacher and student, we directly take proposals produced by the teacher RPN as input for the student R-CNN, realizing one-to-one correspondence. For the RPN stage, since the feature map sizes and pre-set anchors differ between teacher and student, we first transfer the teacher's anchors to the space of student through box transformation. Then we assign

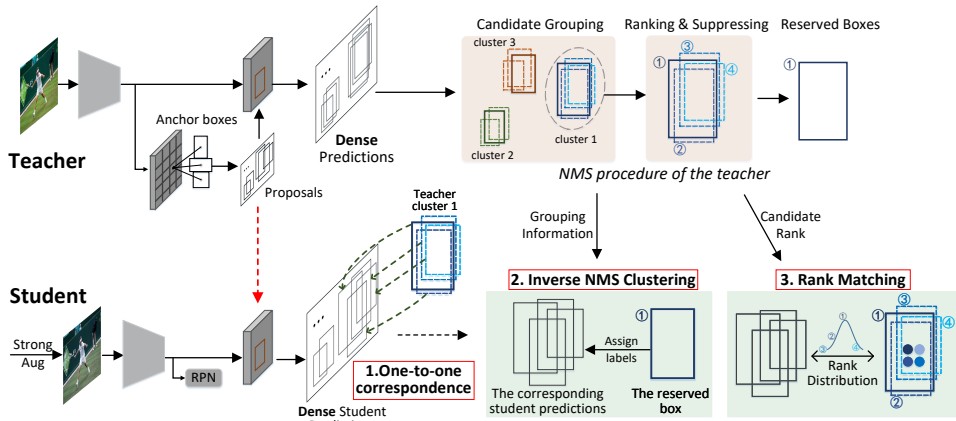

Figure 2: The pipeline of our DTG-SSOD on **unlabeled data**. The training process at R-CNN stage is illustrated, and it can be easily applied to RPN stage with minor modifications. For dense predictions of the teacher, NMS is performed, consisting of candidate grouping, ranking, and suppressing. We design three steps to implement the "dense-to-dense" paradigm. Firstly, one-to-one correspondence between samples of teacher and student is identified. Next, INC assigns training labels for the student based on grouping information revealed in the teacher's NMS. At last, RM regularizes consistency in candidate rank between two models. These three steps are described successively in Sec. 3.2.

the teacher's anchors to the corresponding FPN level based on the anchor box scale. The target FPN level $m_i$ can be obtained by: $m_i = \log_2 \frac{area_i}{S}$, where $area_i$ refers to the area of the teacher anchor box $t_i$, and $S$ is a hyper-parameter and set to 32. Once the target FPN level $m_i$ is obtained, we calculate the Intersection-over-Unions (IoUs) between the teacher anchor $t_i$ and all student anchors $\{s_j\}^{j \in m_i}$ belonging to the FPN level $m_i$, and choose the student anchor with the largest IoU as the corresponding one: $s_i = \arg\max IoU(t_i, s_j), j \in m_i$. Through the proposed matching strategy, one-to-one correspondence between training samples of the teacher and student is identified, making dense behaviour imitation possible.

**Inverse NMS Clustering**. We first revisit the mechanism of NMS by taking RCNN stage as the example. Before NMS, predicted bounding box offsets are applied on proposals to derive regressed boxes. Then a pre-defined score threshold $\tau$ filters out clear background detection boxes. For the remaining detection boxes (or candidates), NMS conducts clustering on them based on their categories and locations. As a result, candidates are grouped into multiple clusters and in each cluster, candidates are supposed to predict the same object. At last, only one candidate with the highest confidence can be retained and others will be suppressed. The process is illustrated in Fig. 2.

To regularize consistency on the NMS behaviour of the teacher and student, we enforce the student to perform the same candidate grouping as the teacher. In the NMS procedure of the teacher, candidates will be grouped into multiple clusters. The cluster indexed by $j$ can be defined as $C_j^t = \{(b_i^t, c_i^t) | 1 \le i \le N_j\}$, where $(b_i, c_i)$ refers to the $i$th candidate, $b$ and $c$ denote the bounding box and category, respectively. $N_j$ is the number of candidates belonging to $C_j$. In each cluster of the teacher (e.g., $C_j^t$), the candidates are highly-overlapped and own the same category $c$ (i.e., $c_1^t = \cdots = c_i^t = c$). Therefore, they are supposed to predict the identical object. Supposing that the $k$th candidate $(b_k^t, c)$ is finally reserved in $C_j^t$, it is the most precise detection box for the object and can behave as the detection target for other candidates in $C_j$. Then the target $(b_k^t, c)$ will inversely assign training labels to the corresponding student samples $\{(b_i^s, c_i^s) | 1 \le i \le N_j\}$. To be specific, the category label $c$ is employed to supervise the classification predictions of the student: $\{c_i^s\}_{i=1}^{N_j}$, and the classification loss can be formulated as:

$$\mathcal{L}_{cls}^u = \frac{\sum_{i=1}^{N_j} f_{cls}(c_i^s, c)}{N_j}, \tag{2}$$

where $f_{cls}$ refers to the Focal loss [16]. Besides, for those teacher candidates, whose scores are lower than the NMS threshold $\tau$, they are clear background boxes, thus their corresponding student samples will obtain background labels. For the regression task, we directly take the reserved box $b_k^t$ as the

regression target for $\{b_i^s\}_{i=1}^{N_j}$. We describe the regression loss as:

$$\mathcal{L}_{reg}^u = \frac{\sum_{i=1}^{N_j} f_{reg}(b_i, b_k^{t'})}{N_j}, \tag{3}$$

where $b_k^{t'}$ is obtained by transforming the $b_k^t$ to the student space; $f_{reg}$ refers to regression loss, e.g., Smooth L1 Loss [17]. There is a clear difference between our method and the traditional sparse-to-dense paradigm. Our method calculates classification and regression labels according to the grouping information revealed in the NMS procedure of the teacher, whilst the sparse-to-dense paradigm employs some hand-crafted label assignment strategies [18, 16, 17] to convert sparse pseudo labels into student's dense training labels, which loses rich teacher guidance.

**Rank Matching.** Through the Inverse NMS Clustering, the student can obtain the same grouping scheme as the teacher. Based on it, we further introduce Rank Matching to align the score rank over clustered candidates between the teacher and student. In each NMS cluster, the candidate boxes are ranked according to the classification scores, then only one candidate will be retained regarding the candidate rank. Therefore, the rank of candidates could contain rich relationship information. In object detection, the candidate rank is validated to be a key element and can affect detection performance much [30, 3, 31, 33]. For example, GFL [30] proposes that taking IoUs between predicted boxes and ground-truths as classification targets contributes to a more precise rank. Different from these methods, our Rank Matching models the score rank over clustered candidates to align the NMS behaviour of the teacher and student. Exactly, we model the score distribution of the teacher and student over the cluster $C_j = \{p_i\}_{i=1}^{N_j}$ as:

$$\mathcal{D}(p_i^t) = \frac{exp(p_i^t/T)}{\sum_{k=1}^{N_j} exp(p_k^t/T)}, \quad D(p_i^s) = \frac{exp(p_i^s/T)}{\sum_{k=1}^{N_j} exp(p_k^s/T)}, \tag{4}$$

where $p_i$ indicates the logit from the target category of the $i$th candidate, and $T$ is the temperature coefficient. The Kullback-Leibler (KL) divergence is adopted to optimize the distance between $\mathcal{D}(p_i^t)$ and $\mathcal{D}(p_i^s)$:

$$\mathcal{L}_{RM} = -\sum_{i=1}^{N_j} D(p_i^t) \log \frac{D(p_i^s)}{D(p_i^t)}. \tag{5}$$

Then $\mathcal{L}_{RM}$ is averaged over all NMS clusters. Rank Matching not only provides relationship information but also enables the student to reserve the identical candidates in each NMS cluster as the teacher. The overall loss function on unlabeled data is defined as:

$$\mathcal{L}^u = \mathcal{L}_{cls}^u + \mathcal{L}_{reg}^u + \beta \mathcal{L}_{RM}^u, \tag{6}$$

where $\beta$ controls contribution of rank matching loss.

## 4 Experiments

### 4.1 Datasets and Evaluation Metric

We benchmark our proposed method on the challenging dataset, MS COCO [34]. Following the convention [28, 9, 13, 10], two training settings are provided, namely **Partially Labeled Data** and **Fully Labeled Data** settings, which validate the method with limited and abundant labeled data, respectively. The val2017 set with 5k images is used as the validation set. We use $AP_{50:95}$ (denoted as mAP) as the evaluation metric. We describe two training settings as follows:

**Partially Labeled Data.** The train2017 set, consisting of 118k labeled images, is used as the training dataset, from which, we randomly sample 1%,2%,5%, and 10% images as labeled data, and set the remaining unsampled images as unlabeled data. Following the practice of previous methods [28, 9, 13], for each labeling ratio, 5 different folds are provided and the final result is the average of these 5 folds.

**Fully Labeled Data.** Here, the train2017 set is employed as the labeled data, and the additional unlabeled2017 set, consisting of 123k unlabeled images, is used as unlabeled data. This setting is to validate SSOD methods can still promote the detector even when labeled data is abundant.

Table 1: Comparison with state-of-the-art methods on COCO benchmark. All results are reported on val2017 set. Under the **Partially Labeled Data** setting, results are the average of all five folds and numbers behind ± indicate the standard deviation. Under the **Fully Labeled Data** setting, numbers in font of the arrow indicate the supervised baseline.

| Methods | Partially Labeled Data | | | | Fully Labeled Data |
| --- | --- | --- | --- | --- | --- |
| | 1% | 2% | 5% | 10% | |
| Supervised Baseline | 12.20±0.29 | 16.53±0.12 | 21.17±0.17 | 26.90±0.08 | 40.9 |
| STAC [28] | 13.97±0.35 | 18.25±0.25 | 24.38±0.12 | 28.64±0.21 | 39.5 $\xrightarrow{-0.3}$ 39.2 |
| Humble Teacher [11] | 16.96±0.35 | 21.74±0.24 | 27.70±0.15 | 31.61±0.28 | 37.6 $\xrightarrow{+4.8}$ 42.4 |
| ISMT [35] | 18.88±0.74 | 22.43±0.56 | 26.37±0.24 | 30.53±0.52 | 37.8 $\xrightarrow{+1.8}$ 39.6 |
| Instant-Teaching [10] | 18.05±0.15 | 22.45±0.15 | 26.75±0.05 | 30.40±0.05 | 37.6 $\xrightarrow{+2.6}$ 40.2 |
| Unbiased Teacher [13] | 20.75±0.12 | 24.30±0.07 | 28.27±0.11 | 31.50±0.10 | 40.2 $\xrightarrow{+1.1}$ 41.3 |
| SED [36] | - | - | 29.01 | 34.02 | 40.2 $\xrightarrow{+3.2}$ 43.4 |
| Rethinking Pse [12] | 19.02±0.25 | 23.34±0.18 | 28.40±0.15 | 32.23±0.14 | 41.0 $\xrightarrow{+2.3}$ 43.3 |
| MUM [15] | **21.88**±0.12 | 24.84±0.10 | 28.52±0.09 | 31.87±0.30 | 37.6 $\xrightarrow{+4.5}$ 42.1 |
| Soft Teacher [9] | 20.46±0.39 | - | 30.74±0.08 | 34.04±0.14 | 40.9 $\xrightarrow{+3.6}$ 44.5 |
| Ours (DTG-SSOD) | 21.27±0.12 | **26.84**±0.25 | **31.90**±0.08 | **35.92**±0.26 | 40.9 $\xrightarrow{\textbf{+4.8}}$ **45.7** |

## 4.2 Implementation Details

Faster RCNN [17] with FPN [37] performs as default detection framework in this paper, and ResNet-50 [38] is adopted as the backbone. Moreover, we use SGD as the optimizer. At the beginning of NMS, a score threshold $\tau$ is leveraged to filter clear background samples. In this work, we set $\tau$ to a relatively high value to exclude noisy teacher samples, whose behaviour is ambiguous and hardly provides helpful guidance. Exactly, we set $\tau$ to 0.9 in NMS of the RPN stage, and 0.45 in NMS of the R-CNN stage, empirically. To stabilize the training, we also warm up the learning on unlabeled data for the first 4k iterations [13]. The training schedules for two training settings are a bit different:

**Partially Labeled Data.** The model is trained for 180k iterations on 8 V100 GPUs with an initial learning rate of 0.01, which is then divided by 10 at 120k iteration and again at 160k iteration. Mini-batch size per GPU is 5, with 1 labeled image and 4 unlabeled images. The loss weight of unlabeled images $\alpha$ is set to 4.0.

**Fully Labeled Data.** The model is trained on 8 GPUs with mini-batch size as 8 per GPU, where 4 labeled images are combined with 4 unlabeled images. We adopt 720k iterations for training, and the initial learning rate is set to 0.01, which is divided by 10 at 480k and 680k iterations. We adopt $\alpha = 2.0$.

## 4.3 Main Results

We compare the proposed DTG-SSOD to the state-of-the-art methods on MS COCO val2017 set in Tab. 1. We first evaluate our method using the **Partially Labeled Data** setting, and experimental results demonstrate our method achieves the state-of-the-art performance under various labeling ratios. Specifically, on 5% and 10% labeling ratios, DTG-SSOD improves the supervised baseline by +10.73 and +9.02 mAP, reaching 31.90 and 35.92 mAP, and it also surpasses the previous best method Soft Teacher [9] by +1.16 and +1.92 mAP, respectively. When the labeling ratio is 2%, our DTG-SSOD yields 26.84 mAP, which is even comparable to the 26.90 mAP supervised baseline using 10% labeled data. When the labeled data is extremely scarce, i.e., 1% labeling ratio, our DTG-SSOD achieves a comparable performance to the previous best method MUM [15], 21.27 vs. 21.88 mAP. MUM [15] designs a strong data augmentation to expand data space, which mixes original image tiles to construct new images. It is worth noting that when the number of labeled data increases (2%,5%, and 10% labeling ratios), our DTD-SSOD outperforms MUM by at least +2 mAP, moreover, our method is orthogonal to these advanced data augmentations [15, 10] and can benefit from them.

As Tab. 1 shows, under the **Fully Labeled Data** setting, our DTG-SSOD surpasses previous methods by a large margin, at least 1.2 mAP. Following the practice of [9, 12], we also apply weak augmentations to labeled data and obtain a strong supervised baseline of 40.9 mAP. Even based on the such strong baseline, DTG-SSOD still attains the biggest improvements of +4.8 mAP, reaching 45.7 mAP, which validates the effectiveness of our method when the amount of labeled data is relatively large.

Table 2: Ablation studies related to effectiveness, efficiency and comparison of two paradigms.

(a) **Effectiveness of Dense Teacher Guidance** in the RPN and RCNN stages.

| RPN | RCNN | mAP | $AP_{50}$ | $AP_{75}$ |
|---|---|---|---|---|
| | | 26.8 | 44.9 | 28.4 |
| ✓ | | 28.5 (+1.7) | 47.7 | 29.5 |
| ✓ | ✓ | **36.3 (+9.5)** | **56.4** | **38.8** |

(b) **Effectiveness of each component of DTG**. Validation is conducted in R-CNN stage.

| INC | | RM | mAP | $AP_{50}$ | $AP_{75}$ |
|---|---|---|---|---|---|
| Cls Label | Reg Label | | | | |
| | | | 28.5 | 47.7 | 29.5 |
| ✓ | | | 32.9(+4.4) | 54.0 | 34.9 |
| ✓ | ✓ | | 35.1(+6.6) | 55.0 | 37.6 |
| ✓ | ✓ | ✓ | **36.3(+7.8)** | **56.4** | **38.8** |

(c) **Comparison of two paradigms** in RPN and R-CNN stages.

| RPN | RCNN | mAP | $AP_{50}$ | $AP_{75}$ |
|---|---|---|---|---|
| spa-den | spa-den | 34.5 | 53.8 | 37.1 |
| spa-den | **den**-den | 35.5 | 55.2 | 38.5 |
| **den**-den | **den**-den | **36.3** | **56.4** | **38.8** |

(d) **Comparison of two paradigms** under various labeling ratios.

| Methods | 1% | 5% | 10% |
|---|---|---|---|
| sparse-dense | 21.1 | 31.1 | 34.5 |
| dense-dense | **21.3** | **31.9** | **36.3** |

(e) Study on **training efficiency**. 90k is iterations.

| Methods | 10% ratio | | 100% ratio | |
|---|---|---|---|---|
| | 90k | 180k | 360k | 720k |
| Soft Teacher [9] | - | 34.0 | - | 44.5 |
| Our DTG-SSOD | 33.7 | **36.3** | 44.7 | **45.7** |

Table 3: Studies on hyper-parameters and generalization. Sup Baseline denotes supervised baseline.

(a) Study on **temperature T**.

| T | mAP | $AP_{50}$ | $AP_{75}$ |
|---|---|---|---|
| 0.5 | 35.6 | 55.1 | 38.4 |
| 1.0 | **35.8** | **55.5** | **38.5** |
| 5.0 | 35.6 | 55.3 | 38.4 |

(b) Study on **loss weight** $\beta$

| $\beta$ | mAP | $AP_{50}$ | $AP_{75}$ |
|---|---|---|---|
| 1.0 | 35.8 | 55.5 | 38.5 |
| 3.0 | **36.3** | **56.4** | **38.8** |
| 5.0 | 36.1 | 55.9 | 38.8 |

(c) Experiment on one-stage detectors.

| Methods | mAP | $AP_{50}$ | $AP_{75}$ |
|---|---|---|---|
| Sup Baseline | 28.1 | 43.9 | 29.4 |
| sparse-dense | 35.8 | 53.3 | 38.3 |
| dense-dense | **37.1** | **55.6** | **40.4** |

## 4.4 Ablation Experiments

In this section, we conduct extensive ablation studies to validate the key designs. Unless specified, all experiments adopt a single data fold of the 10% labeling ratio as the training dataset.

**Effects of Dense Teacher Guidance.** Instead of conventional pseudo boxes with category labels provided, our method offers dense supervisory signals. We first validate that Dense Teacher Guidance (DTG) is a feasible learning paradigm for SSOD in Tab. 2(a), where we apply DTG to RPN and R-CNN stages step by step. On unlabeled data, when only RPN is optimized with the DTG and R-CNN is excluded from training, it can bring 1.7 mAP gains to the supervised baseline. Furthermore, we apply DTG to both RPN and RCNN, and it yields 36.3 mAP, outperforming the baseline by +9.5 mAP. Significant improvements against the supervised baseline indicate dense teacher guidance can behave as effective supervision signals for learning on unlabeled data. DTG consists of two key components, i.e., Inverse NMS Clustering (INC) and Rank Matching (RM), then we delve into these two parts with results reported in Tab. 2(b). Through imitating the candidate grouping of the teacher in NMS, INC provides appropriate training labels for the student. With the derived classification label, the model enjoys 4.4 mAP gains. When regression labels are also applied, another 2.2 mAP improvement is found, reaching 35.1 mAP. Based on INC, RM is introduced to provide teacher's score rank over clustered candidates for the student, which enables the student to reserve the same candidate in each NMS cluster as the teacher. As results show, RM can bring an obvious improvement, +1.2 mAP, demonstrating comprehensive alignment on NMS behaviour is beneficial. Besides, the gains also indicate the candidate rank carries helpful relation information learned by the teacher.

**Comparison of sparse-to-dense and dense-to-dense paradigms.** In this work, we replace the traditional sparse-to-dense (spa-den for short) paradigm with new dense-to-dense (den-den) paradigm. Detailed comparisons are conducted in Tab. 2(c-d). We first build a sparse-to-dense baseline, then incrementally substitute our den-den paradigm into each network stage (i.e., RPN, RCNN) in Tab. 2(c). Thanks to the appropriate mixing ratio $r = 1/4$ between labeled and unlabeled images and Focal Loss, our sparse-to-dense baseline achieves comparable performance (34.5 mAP) to the previous methods. When we substitute the den-den paradigm into R-CNN, it yields 35.5 mAP, which is 1.0 points higher than the spa-den R-CNN. After that, we further apply den-den to RPN, which attains +0.8 mAP gains. These experiments validate the proposed dense-to-dense paradigm is superior to the traditional sparse-to-dense paradigm in both RPN and RCNN stages.

On the other hand, we also compare two paradigms under various labeling ratios in Tab. 2(d). Under 5% and 10% ratios, our method is obviously better than the sparse-to-dense baseline, and

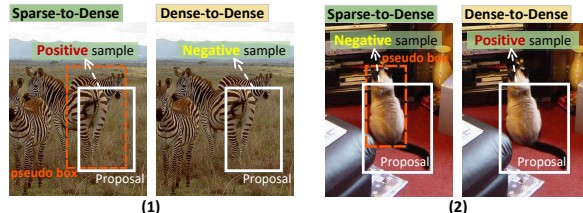 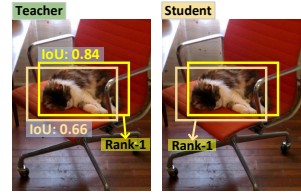

Figure 3: For the *identical* proposals (white), two paradigms will assign *different* training labels. The **poor** and **precise** proposals are displayed in sub-fig (1) and (2), respectively.

Figure 4: One example to illustrate how candidates are ranked by the teacher and student.

surpasses it by 0.8% and 1.8% mAP, respectively. Nevertheless, when the labeling ratio is 1%, the performance gap shrinks. This is easy to understand: When labeled data is extremely scarce, detection performance is poor. Thus the predictions of the teacher are relatively inaccurate, resulting in noisy teacher guidance. Especially for Rank Matching, the teacher model fails in modeling accurate candidate relations due to weak performance. To extract accurate guidance from the very weak teacher model is left for future work.

**Training efficiency.** Apart from the standard training schedule, we also attempt to halve the training iterations in Tab. 2(e). Under the 10% labeling ratio, our DTG-SSOD yields 33.7 mAP with only 90k iterations, which is very close to the previous best method Soft Teacher with 180k iterations, 34.0 mAP. Besides, under the Fully Labeled Data setting (100% labeling ratio), our method halves the training iterations of Soft Teacher and achieves even better performance, 44.7 vs. 44.5 mAP. These results validate the high efficiency of DTG-SSOD.

**Ablation study on hyper-parameters.** There are two hyper-parameters (i.e., $T$ and $\beta$) in Rank Matching, where temperature $T$ is used to control the flatness of score distribution over clustered candidates, and $\beta$ performs as the loss weight of $\mathcal{L}_{RM}$. In Tab 3(a), we ablate $T$ with $\beta$ set to 1.0. As results indicate, the performance of RM is insensitive to $T$, and we adopt $T = 1.0$ for the following experiments. A coarse search on $\beta$ is shown in Tab 3(b), where $\beta = 3.0$ achieves the best performance.

**Experiments on one-stage detectors.** We also conduct experiments to verify the proposed dense-to-dense paradigm can generalize to one-stage detectors well. We take the GFL [30] as the default one-stage detector, because of its high efficiency and strong performance. Comparisons between the sparse-to-dense with dense-to-dense paradigms are reported in Tab 3(c). As results indicate, the dense-to-dense paradigm surpasses the counterpart by +1.3 mAP, demonstrating the proposed dense teacher guidance can work well on one-stage detectors.

### 4.5 Analyses

In this section, we provide some analyses on why dense teacher guidance is superior to sparse pseudo labels. In the sparse-to-dense paradigm, some pseudo boxes are poorly localized, which will mislead the standard IoU-based label assignment and cause noisy training labels for the student. Without the dependency on pseudo boxes, our method can effectively alleviate this problem. Two examples are shown in Fig. 3, where the *identical* student proposals will obtain *different* training labels in the two paradigms. Compared with the sparse-to-dense paradigm, our method obviously offers more precise training labels for the student. In the sub-fig (1), the sparse-to-dense paradigm treats the poor proposal (in white) as a positive sample. Because the proposal has a high IoU value (i.e., 0.6) with the coarse pseudo box (in red), therefore, it reaches the requirement of positive samples in the IoU-based label assignment. In contrast, our Inverse NMS Clustering takes teacher predictions into consideration and succeeds in suppressing this misleading positive. Concretely, the teacher model predicts low confidence (i.e., 0.3) for this proposal, thus it will be a clear background sample in NMS. Another example is shown in the sub-fig (2). Due to the poor localization of the pseudo box, a relatively **precise** proposal is wrongly assigned as a negative sample in the sparse-to-dense paradigm. However, our INC can avoid this problem by taking the teacher's NMS behavior into consideration.

In Fig. 4, we also provide an example to illustrate that the teacher is better at modeling candidate rank. Between the two candidates shown in the figure, the teacher ranks the one with more precise

localization first, while the student fails. Therefore, distilling the capacity of modeling candidate rank from the teacher to student is beneficial, which is missed in sparse pseudo labels.

## 5   Conclusion

In this work, we analyze the traditional semi-supervised object detection methods, where *sparse* pseudo boxes with the corresponding category labels are adopted for *dense* supervision of the student, formulating the popular "sparse-to-dense" paradigm. We point out that the sparse-to-dense paradigm could accumulate noise in the hand-crafted NMS and label assignment process, and further lack dense and direct teacher guidance. To address the problem, we propose a novel "dense-to-dense" paradigm, which integrates Dense Teacher Guidance into the student training. Specifically, we introduce Inverse NMS Clustering and Rank Matching to instantiate the dense guidance: INC leads the student to perform the same candidate grouping in NMS as the teacher does; RM passes the teacher's knowledge of rank distribution over clustered candidates to the student. Through INC and RM, the student can receive sufficient, informative, and dense guidance from the teacher, which naturally leads to better performance. On COCO benchmark, extensive experiments under various labeling ratios validate that our DTG-SSOD can achieve SOTA performance in both accuracy and efficiency.

## Acknowledgments and Disclosure of Funding

This work is partially supported by National Natural Science Foundation of China (Grant No. 62172225), the Fundamental Research Funds for the Central Universities (No. 30920032201), and the Young Scientists Fund of the National Natural Science Foundation of China (Grant No.62206134).

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
