# A    Experiments on PASCAL VOC

## A.1    Dataset and Implementation Details

**PASCAL VOC**. We also conduct experiments on PASCAL VOC for a comprehensive comparison with previous methods. Following [15, 13, 28, 10], we adopt two training settings: **VOC12** and **VOC12+COCO20cls**. For the **VOC12** setting, we use the VOC07 trainval set as the labeled training data, and the VOC12 trainval set as the unlabeled training data; For the **VOC12+COCO20cls** setting, apart from the VOC12 trainval set, we also employ COCO20cls as unlabeled data, where images are sampled from the COCO training set [34] to only contain 20 PASCAL VOC classes. All experiments are tested on the VOC07 test set. We adopt $AP_{50}$ and $AP_{50:95}$ as the evaluation metrics.

**Implementation Details**. All models are trained on 8 V100 GPUs for 90K iterations with 5 images per GPU. The initial learning rate is set to 0.01, then divided by 10 at 60k and 80k iterations. The mixing ratio $r$ between labeled and unlabeled images is set to 1:4. We adopt $\alpha = 4$ to balance the losses on labeled and unlabeled images.

## A.2    Experimental Results

In Tab. 4, we compare our method with state-of-the-art methods on PASCAL VOC. Using **VOC12** as the unlabeled data, our method surpasses previous methods by a large margin, especially in $AP_{50:95}$, our DTG-SSOD surpasses the previous best method MUM [15] by +2.96 points. Our results also hold on the larger amount of unlabeled images, specifically, when the combination of VOC12 and COCO20cls is used as unlabeled images, our method attains further improvements, reaching 80.63 $AP_{50}$ and 54.58 $AP_{50:95}$. Since our DTG-SSOD enforces the student to mimic the teacher's behaviour revealed in the NMS process, our detector can precisely model relations between different candidates and reserve high-quality detection boxes as final results, resulting in higher $AP_{50:95}$.

Table 4: Comparison with state-of-the-art methods on PASCAL VOC. Both settings adopt the VOC07 trainval set as the labeled data.

| Methods | VOC12 | | VOC12+COCO20cls | |
|---|---|---|---|---|
| | $AP_{50}$ | $AP_{50:95}$ | $AP_{50}$ | $AP_{50:95}$ |
| Supervised Baseline | 69.71 | 42.49 | 69.71 | 42.49 |
| CSD [27] | 74.70 | - | 75.10 | - |
| STAC [28] | 77.45 | 44.64 | 79.08 | 46.01 |
| Instant-Teaching [10] | 78.30 | 48.70 | 79.00 | 49.70 |
| ISMT [35] | 77.23 | 46.23 | 77.75 | 49.59 |
| Unbiased Teacher [13] | 77.37 | 48.69 | 78.82 | 50.34 |
| MUM [15] | 78.94 | 50.22 | 80.45 | 52.31 |
| Ours (DTG-SSOD) | **79.20** | **53.18** | **80.63** | **54.58** |

# B    Additional Ablation Study

At the beginning of NMS, a score threshold $\tau$ is set to filter out clear background samples. In our DTG-SSOD, we assume only teacher candidates with higher scores than $\tau$ can provide precise and informative knowledge for the student training; in contrast, for those teacher candidates, whose scores are lower than $\tau$, we assign background labels to their corresponding student samples. In Tab. 5, we conduct ablation studies on NMS threshold $\tau$ for the R-CNN stage. The best performance is achieved when the threshold $\tau$ is set to 0.45.

Table 5: Study on **NMS threshold** $\tau$

| $\tau$ | mAP | $AP_{50}$ | $AP_{75}$ |
|---|---|---|---|
| 0.40 | 35.8 | 55.9 | 38.6 |
| 0.45 | **36.3** | **56.4** | **38.8** |
| 0.50 | 33.9 | 53.1 | 36.3 |

# C    Additional Analyses

We further provide some additional examples to analyze the advantage of our DTG-SSOD over the traditional sparse-to-dense paradigm. We conclude that the superiority of our method comes from two aspects:

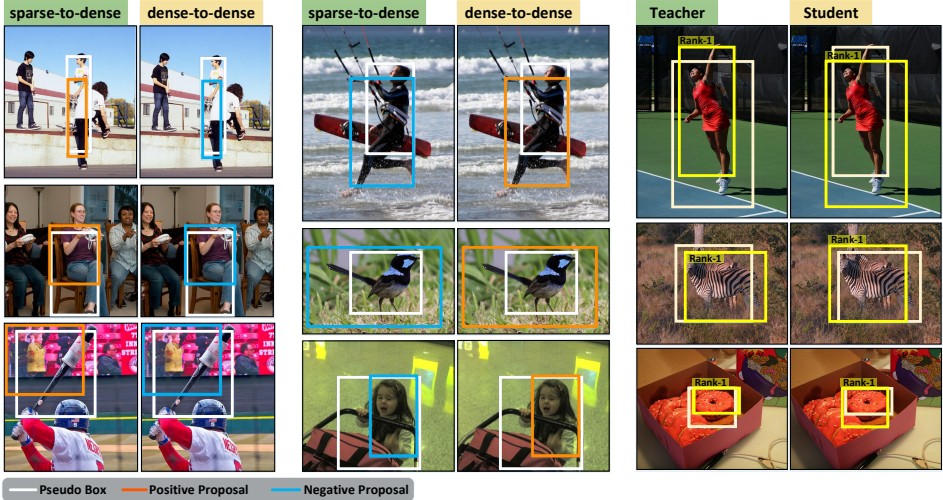

**(a) Suppress misleading positives**  **(b) Address ambiguous samples**  **(c) The teacher can reserve precise boxes**

Figure 5: Some visualized examples to demonstrate the advantage of our method over the traditional sparse-to-dense paradigm. (a-b) For the same student proposals, our dense-to-dense paradigm and traditional sparse-to-dense paradigm will assign different labels. It is obvious that our dense-to-dense paradigm can assign more precise and reasonable training labels. (c) The teacher is better at modeling relations over cluster candidates than the student.

**The DTG-SSOD can provide precise and reasonable training labels for the student.** Due to coarse and inaccurate pseudo boxes, the hand-crafted label assignment methods confuse the decision boundary of the positive and negative training samples. In Fig. 5(a), some poorly localized proposals are assigned as positive ones in the sparse-to-dense paradigm, however, our method can successfully suppress these misleading labels. Specifically, in our DTG-SSOD, the teacher model predicts low scores ($< \tau$) for these proposals, thus they are considered as clear background samples. In Fig. 5(b), for these ambiguous proposals, which partially detect salient regions of objects, our method assigns positive labels and applies regression supervision to them. As a result, these proposals can be suppressed by NMS and avoid duplicated detections.

**Relationship information offered by Rank Matching promotes the student training.** In Fig. 5(c), the teacher model can assign higher scores to high-quality candidates, thereby reserving more precise boxes in NMS. Therefore, relationship information learned by the teacher is beneficial to the student, and it does not exist in previous methods.

## D  Details about Data Augmentation

We summary transformations used in the weak and strong augmentation in Fig. 6.

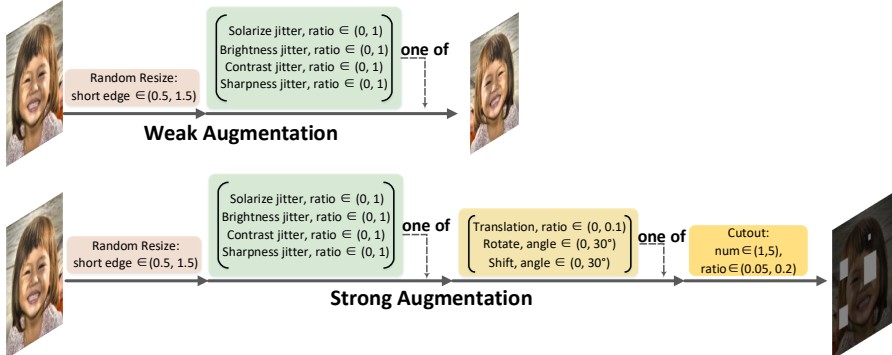

Figure 6: The summary of transformations used in weak and strong augmentation.

# E   Broader Impact

Object detection is a fundamental computer vision task and beneficial to a wide range of applications, including autonomous driving, smart surveillance, and robotics. This work aims to use easily accessible unlabeled data to promote object detection and reduce dependency on hand-labeled data. Any object detection application can benefit from our work. We do not foresee obvious undesirable ethical/social impacts at this moment.