# OpenReview forum: "DTG-SSOD: Dense Teacher Guidance for Semi-Supervised Object Detection"
_NeurIPS.cc/2022/Conference — NeurIPS 2022 Accept_

### Official Review · Reviewer_YaeS · 2022-07-11

**Rating:** 5
**Confidence:** 4
**Soundness:** 3 good
**Presentation:** 3 good
**Contribution:** 3 good

**Summary:**

This paper introduces a new dense-to-dense” distillation to supervise the training of object detectors. The authors consider the advantages of the dense-to-dense paradigm and point out that the sparse-to-dense paradigm could accumulate noise in the hand-crafted NMS and label assignment process. They propose the Inverse NMS Clustering (INC) and Rank Matching (RM) to instantiate the dense supervision. Finally, the method achieve SOTA performance in both accuracy and efficiency.

**Questions:**

1. Which is better between your method and the feature distillation(distillation on the final layer)?
2. The supervision in Inverse NMS Clustering is a "sparse-to-dense" paradigm, which is supervised by a reserved box. The author claims that the weakness of the "sparse-to-dense" at the beginning.

**Limitations:**

1. The questions above confuse me. Q1,Q2 .
2. The method is not very interesting, but the paper presents extensive and insightful ablations study about their methods.

**Strengths And Weaknesses:**

- The central insight of the work is simple and intuitive i.e. using dense supervision can be more informative regarding the information loss of NMS and score filtering. And the authors propose some solutions(INC and RM) to these problems.
- The paper presents extensive and insightful ablations showing the importance of various choices (e.g. temperature T, choice of target boxes in INC etc.), and overall, all the choices made are empirically well-justified.

---

> ### Author Response · Authors · 2022-08-02
> **Response to Reviewer YaeS**
>
> Thanks for your constructive comments. We respond to them below.
>
> **Q1. Which is better between your method and the feature distillation (distillation on the final layer)?**
>
> **A1.** We explain the differences between our method and vanilla distillation methods as follows:
>
> 1. Vanilla distillation methods directly take dense predictions of the teacher as knowledge to distill without any adaptation, in contrast, our method converts the teacher's dense predictions into the candidate grouping information and candidate rank, which serve as more informative and instructive knowledge for  student training. **Some important properties of object detection are considered in our method**. For example, considering that accurate box rankings benefit the NMS processing [3, 28], we propose Rank Matching to regularize consistency in candidate rank between the teacher and student. On the contrary, vanilla distillation methods are originally designed for the image classification task, and directly transferring them to object detection ignores the fact that object detection has a more complicated mechanism than classification. Lacking essential adaptations, the vanilla distillation methods may only obtain sub-optimal distillation performance in object detection.
>
> 2. **Vanilla distillation methods hardly adapt to semi-supervised settings well.** In the Mean-Teacher scheme, distinct data augmentations are applied to the teacher and student, which increases feature discrepancies between the teacher and student, as a result, feature distillation (including distillation on FPN features and R-CNN features) is hard to optimize. On the other hand, distillation on the final layer takes the predictions of the teacher as soft labels. However, previous works (e.g., FixMatch [25]) validated that the hard pseudo labels with low entropy can perform favorably against the soft labels on unlabeled data. Neither feature distillation nor final layer distillation can perform well in  semi-supervised settings. In contrast, our method is tailored for semi-supervised settings.
>
> We conduct expriments to verify our analysis. Experimental results are listed as follows:
>
> | methods                    | mAP      | AP$_{50}$ | AP$_{75}$ |
> | -------------------------- | -------- | --------- | --------- |
> | supervised baseline        | 26.8     | 44.9      | 28.4      |
> | FPN feature distillation   | 28.7     | 47.9      | 30.1      |
> | R-CNN feature distillation | 29.2     | 48.2      | 30.6      |
> | final layer distillation   | 33.3     | 52.9      | 35.8      |
> | DTG-SSOD (ours)            | **36.3** | **56.4**  | **38.8**  |
>
> Experiments are conducted under the 10% labeling ratio. As the results show, our method surpasses other distillation methods by a large margin, at least **3.0 mAP**.  These results support our analyses.
>
> **Q2. The supervision in Inverse NMS Clustering is a "sparse-to-dense" paradigm, which is supervised by a reserved box. The author claims that the weakness of the "sparse-to-dense" at the beginning.**
>
> **A2.** In the Inverse NMS Clustering (INC), **not only reserved boxes but also dense group information** are employed to supervise the student training. INC is proposed to lead the student to group candidate boxes into clusters in NMS as the teacher does, by learning the **grouping information** revealed in the teacher's NMS procedure. Grouping information provides NMS status for each candidate box, indicating which cluster each candidate belongs to. Obviously, group information serves as dense supervision, enabling  INC to be a "dense-to-dense" paradigm.

---

> > ### Comment · Reviewer_YaeS · 2022-08-09
> > **Acknowledge the rebuttals**
> >
> > Thanks for your rebuttal. The author provides detailed experiments to prove this. It has addressed my concerns in the rebuttal. So I think it is a good paper.

---

### Official Review · Reviewer_CxZW · 2022-07-12

**Rating:** 7
**Confidence:** 3
**Soundness:** 3 good
**Presentation:** 2 fair
**Contribution:** 3 good

**Summary:**

The authors propose a novel SSOD method that guides the student network to behave like the teacher network's NMS. The proposed method enforces a dense-to-dense training signal instead of the more common sparse-to-dense strategy in other SSOD methods. The proposed method is composed of two parts: (1) INC which leads the student to group candidate boxes into clusters in NMS as the teacher does, and (2) RM which imitates the rank distribution of the teacher over clustered candidates.

The author demonstrates the effectiveness of the proposed method compared to SoTA SSOD methods. They also conduct thorough ablations and analyses to help understand the contribution of each component and the advantage of the proposed method over the sparse-to-dense strategy.

**Questions:**

Please see the questions in the weakness part of the "Strengths And Weaknesses" section.

**Limitations:**

In lines 148-149, the authors claim, "In theory, our SSOD method is independent of the detection framework and can be applicable to both one-stage and two-stage detectors." However, as Unbiased Teacher v2 [1] pointed out, it is non-trivial to generalize a method developed on a two-stage detector to a one-stage one.

[1] Liu, Yen-Cheng, Chih-Yao Ma, and Zsolt Kira. "Unbiased Teacher v2: Semi-Supervised Object Detection for Anchor-Free and Anchor-Based Detectors." Proceedings of the IEEE/CVF Conference on Computer Vision and Pattern Recognition. 2022.

**Strengths And Weaknesses:**

### Strength

1. Enforcing the student to mimic the teacher's NMS behavior is somewhat novel in SSOD.

1. This work is well motivated for why using dense-to-dense is a preferred training strategy.

1. The authors conduct thorough ablations (Sec.4.4) and analyses (Sec.4.5) for each component of the proposed method as well as why the proposed dense-to-dense method works better than the sparse-to-dense methods.

1. As shown in Tab.2(b), It is surprising that including the box regression loss is beneficial for the final performance without any tailored method or tricks.

### Weakness

1. It would be good to verify whether (or to what degree) the student network actually mimics the teacher's NMS behavior.

1. In Eq.4, is $p^t_i$ probability or logit? It seems weird to pass the probability through exponential and softmax. Also, is $p^t_i$ the output from an additional fc/mlp head or just the output from the existing objectiveness/classification head of the detector?

1. Is RM applied to the R-CNN part or the RPN part as well?

1. In Tab.2, what is the setting here? Eg. the amount of labeled data. Could the findings generalize to other settings (eg. different amounts of labeled data)? In Tab.2(a), it makes more sense to compare with sparse-to-dense rather than supervised baseline. In Tabl.2(e), it is better presented as a figure containing the val curve of both conditions.

1. It would be helpful to also include the results of the sparse-to-dense counterpart in Tab.1.

1. Are Fig.3 cherry-picked results or randomly selected results? It would be good to also include the failure modes and provide analyses and discussions.

1. In Fig.1(b), Inverse NMS Clustering, what does that number of star and triangle shapes mean? Also, why do the student samples not contain the reserved box?

---

> ### Author Response · Authors · 2022-08-02
> **Response to Reviewer CxZW**
>
> Thanks for your constructive comments. We respond to them below.
>
> **Q1. Verifying whether (or to what degree) the student network actually mimics the teacher's NMS behavior.**
>
> **A1.** We define a metric, termed *Overlap Ratio (OR)*, to measure the similarity between NMS behavior of two models. Inspired by the definition of IoU, we formulate the OR as:
> \begin{equation}
>     OR = \frac{|box_{s} \cap box_{t}|}{|box_{s} \cup box_{t}|},
> \end{equation}
> where $box_{s}$ and $box_{t}$ refer to reserved boxes after the NMS for the student and teacher; $|box_{s} \cap box_{t}|$ denotes the number of overlapped boxes between $box_{s}$ and $box_{t}$. Only when $box_{s}$ and $box_{t}$ **originate from the same proposal**, they are considered as overlapped. Moreover, $|box_{s} \cup box_{t}| = |box_{s}| + |box_{t}| - |box_{s} \cap box_{t}|$, where $|box_{s}|$ denotes the number of  $box_{s}$.
> Higher OR values indicate more similar NMS behavior between the teacher and student. Next, we calculate the OR for the DTG-SSOD and sparse-to-dense baseline. Two checkpoints with similar performance are adopted from two paradigms respectively for analysis, and analyses are conducted on coco val2017 set. The OR of the sparse-to-dense baseline is **33.4%**, and our DTG-SSOD surpasses it by **4.1 points**, reaching **37.5%**. The significant improvements in OR validate that the discrepancy between NMS behavior of the teacher and student is narrowed through the proposed INC and RM.
>
> **Q2. In Eq.4, is $p_{i}^{t}$ probability or logit? Also, is $p_{i}^{t}$ the output from an additional fc/mlp head or just the output from the existing objectiveness/classification head of the detector?**
>
> **A2.** In Eq.4, $p_{i}^{t}$ is the logit. We correct the descriptions about Eq.4 in the revision. $p_{i}^{t}$ is just the output from the existing classification head of the detector.
>
> **Q3.** **Is RM applied to the R-CNN part or the RPN part as well?**
>
> **A3.** Rank Matching is also applied to the RPN part. From Tab.2(c), applying INC and RM to RPN brings a gain of + 0.8 mAP against the sparse-to-dense RPN.
>
> **Q4. What's the setting in Tab.2?  In Tab.2(a), it makes more sense to compare with sparse-to-dense rather than supervised baseline. In Tab.2(e), it is better presented as a figure containing the val curve of both conditions.**
>
> **A4.** As stated in Line 280-281, unless specified, all ablation studies in Tab.2 adopt a single data fold of the **10% labeling ratio** as the training data. In Tab.2(d) and Line 309-316, we discuss the generalization of our method on various labeling ratios. The experimental results indicate our method consistently performs favorably against the traditional sparse pseudo labels under various settings.
>
> Thank you for your kind and professional suggestions on Tab.2(a) and Tab.2(e). A comparison with the sparse-to-dense is already listed in Tab.2(c), thus we report the comparison with the supervised baseline in Tab.2(a). We will substitute the val curve for Tab.2(c), once we obtain the complete val curve of Soft Teacher.
>
> **Q5. It would be helpful to also include the results of the sparse-to-dense counterpart in Tab.1.**
>
> **A5.** Good suggestion! Most of the results in Tab.1 are averaged on 5 different data folds, which is time-consuming. We hardly finish all experiments on the sparse-to-dense counterpart before the rebuttal deadline. However, we will add it in the final version.
>
> **Q6. Are Fig.3 cherry-picked results or randomly selected results?**
>
> **A6.** Qualitative results in Fig.3 are randomly selected. More results are attached in *supplementary material*. We look up many visualized examples and only few valuable failure cases are observed. In most of the examples, our DTG-SSOD consistently offers more precise supervision for the student than the sparse-to-dense baseline.
>
> **Q7. In Fig.2(b), Inverse NMS Clustering, what does that number of star and triangle shapes mean? Also, why do the student samples not contain the reserved box?**
>
> **A7.** In Fig.2(b), there is no special meanings for the number of stars and triangles. We should have kept their number the same to avoid ambiguity. Moreover, the reserved box should be contained in the student samples, which is wrongly illustrated in the original figure. For a better presentation, we update Figure 2 in the revision. Please refer to it for a better understanding.

---

> > ### Author Response · Authors · 2022-08-02
> > **Response to Reviewer CxZW**
> >
> > **Q8. It is non-trivial to generalize a method developed on a two-stage detector to a one-stage one.**
> >
> > **A8.** Our DTG-SSOD can perform well on most anchor-based detectors, regardless of one-stage or two-stage ones. We take Generalized Focal Loss (GFL) [28], a popular one-stage detector, as an example, to demonstrate the generalization of DTG-SSOD. The results are listed as follows:
> >
> > | detector | method                   | mAP      | AP$_{50}$ | AP$_{75}$ |
> > | -------- | ------------------------ | -------- | --------- | --------- |
> > | GFL      | supervised baseline      | 28.1     | 43.9      | 29.4      |
> > | GFL      | Sparse-to-Dense baseline | 35.8     | 53.3      | 38.3      |
> > | GFL      | DTG-SSOD (ours)          | **37.1** | **55.6**  | **40.4**  |
> >
> > Experiments are conducted under the 10% labeling ratio. On the GFL, our method surpasses the sparse-to-dense counterpart by +1.3 mAP. However, generalizing the proposed method to anchor-free detectors may be no-trivial, and we leave it for future work.

---

> ### Comment · Reviewer_CxZW · 2022-08-07
> **Comments**
>
> Thanks for the rebuttal. The authors have properly addressed the reviewer's questions in the rebuttal. Thus, the reviewer decided to keep the original rating and suggest accepting this paper.

---

### Official Review · Reviewer_K4Pn · 2022-07-13

**Rating:** 6
**Confidence:** 4
**Soundness:** 3 good
**Presentation:** 3 good
**Contribution:** 3 good

**Summary:**

This paper focuses on the pseudo-labeling strategy in semi-supervised object detection. Unlike previous approaches which generate sparse pseudo labels from teacher detector for each image and match these sparse pseudo labels to dense proposal and object detection predictions from student detector, this paper proposes a "dense-to-dense" paradigm. Inverse NMS Clustering and Rank Matching components are proposed to instantiate the dense-to-dense paradigm. Experiments on COCO show that the proposed approach obtains the state-of-the-art semi-supervised object detection results.

**Questions:**

How could the proposed approach be applied to other object detectors, especially the detectors without dense RPN/proposal-wise predictions and NMS (e.g., DETR)?

**Limitations:**

It's unclear how it can be applied to recent end-to-end object detectors (e.g., DETR) which don't have dense RPN and proposal-wise predictions.

**Strengths And Weaknesses:**

### Strengths
- The proposed approach is interesting.
- Most parts of the paper are well written and easy to understand.
- Very promising results are obtained by the proposed approach and detailed ablation experiments are conducted.

### Weaknesses
- Some parts of the paper writing can be improved. The Figure 2 is not very clear about how inverse NMS and Rank Matching are performed.
- The proposed approach is mainly applied to Faster R-CNN based detector. It's unclear how it can be applied to recent end-to-end object detectors (e.g., DETR) which don't have dense RPN and proposal-wise predictions.

---

> ### Author Response · Authors · 2022-08-02
> **Response to Reviewer K4Pn**
>
> Thanks for your constructive comments. We respond to them below.
>
> **Q1. Some parts of the paper writing can be improved. Figure 2 is not very clear.**
>
> **A1.**  In the revision, we have refined all three figures and polished the Analyses part (Sec. 4.5) for a better presentation.
>
> **In the new Figure 2**, we make the following modifications:
>
> 1. We elaborately describe the NMS procedure of the teacher, including candidate grouping, ranking, and suppressing. It is underlined that dense teacher guidance is extracted from the teacher's NMS process.
> 2. We split the proposed dense-to-dense paradigm into three steps. At first, one-to-one correspondence between samples of teacher and student is identified, getting ready for the following behavior imitation. Then, Inverse NMS Clustering is performed. The grouping information and reserved boxes offered by the teacher's NMS procedure are converted into dense training labels for student samples. Finally, rank distributions over clustered candidates are enforced to be consistent between the teacher and student.
>
> **Q2. How could the proposed approach be applied to other object detectors?**
>
> **A2.** The principle of our DTG-SSOD is to employ powerful direct, dense teacher supervision. For CNN-based detectors (e.g., Faster R-CNN), we instantiate the dense supervision with NMS behavior of dense predicted boxes, achieving favorable performance. However, for query-based detectors (e.g., DETR) without dense predictions and NMS, we can instantiate dense supervision with **bipartite graph matching of the teacher**. Considering that object queries are relatively dense (e.g., 900 queries are used in DINO [R-1]), the bipartite matching results for object queries can be regarded as dense information. Moreover, recent works [R-1, R-2] validate that the slow convergence of DETR results from the instability of bipartite graph matching. Thanks to stronger performance, the teacher model is supposed to have more stable bipartite matching than the student. Based on these insights, predicted bipartite matching of the teacher is able to serve as dense knowledge to provide consistent optimization goals for the student and stabilize the student training.
>
> We adopt the two-stage detector (i.e., Faster R-CNN) as the default detection framework in the paper. Here, we also apply our DTG-SSOD to a popular **one-stage detector**, Generalized Focal Loss (GFL) [28], to demonstrate the generalization. The results are listed as follows:
>
> | detector | method                   | mAP      | AP$_{50}$ | AP$_{75}$ |
> | -------- | ------------------------ | -------- | --------- | --------- |
> | GFL      | supervised baseline      | 28.1     | 43.9      | 29.4      |
> | GFL      | Sparse-to-Dense baseline | 35.8 | 53.3      | 38.3      |
> | GFL      | DTG-SSOD (ours)          | **37.1** | **55.6**  | **40.4**  |
>
> Experiments are conducted under the 10% labeling ratio. From the table, our method surpasses the sparse-to-dense baseline by **+1.3 mAP**, validating that our method can perform well on both two-stage and one-stage detectors.
>
> #### **References**
>
> [R-1] Zhang, Hao, et al. "Dino: Detr with improved denoising anchor boxes for end-to-end object detection." arXiv preprint arXiv:2203.03605 (2022).
>
> [R-2] Li, Feng, et al. "Dn-detr: Accelerate detr training by introducing query denoising." Proceedings of the IEEE/CVF Conference on Computer Vision and Pattern Recognition. 2022.

---

### Official Review · Reviewer_2EWD · 2022-07-19

**Rating:** 6
**Confidence:** 4
**Soundness:** 3 good
**Presentation:** 2 fair
**Contribution:** 3 good

**Summary:**

This paper proposes a "dense-to-dense" paradigm that utilizes the dense guidance of teacher to supervise the students. Specifically, Dense Teacher Guidance (DTG)'s dense supervision is achieved by Inverse NMS Clustering (INC) and Rank Matching (RM), which regularizes the consistency on NMS between the teacher and student. INC leads the student to group the candidate boxes into clusters in NMS as the teacher does, so that the student obtains the same grouping scheme of NMS with the teacher. Rank Matching is further introduced to align the score rank over the clustered candidates between teacher and student.

**Questions:**

Could you better illustrate Figure 3? It is poorly presented.

**Limitations:**

Despite fair technical merit and solid experimental validation, the visual illustration in this paper is pretty poor.

**Strengths And Weaknesses:**

Strength:
1. The motivation of "dense-to-dense" paradigm is strong and clear. The limitation of "sparse-to-dense" is well presented.
2. The proposed Inverse NMS Clustering (INC) and Rank Matching are novel. The rank of the samples within each cluster learned by the teacher serves as informative dense supervision, which enables the student to reserve the same candidates as the teacher during NMS.
3. Ablation studies show the effectiveness of each proposed components or strategies.
Weakness:
1. Figure1 and 2 are not illustrative. These two figures fail to give an intuitively visual demonstration on how DTG-SSOD works.
2. Table 1 is expected to show more results under different labeling ratios, e.g., 20%, 30%, 50%.
3. Figure 3 is not illustrative. I don't understand it.

---

> ### Author Response · Authors · 2022-08-02
> **Response to Reviewer 2EWD**
>
> Thanks for your constructive comments. We respond to them below.
>
> **Q1. Figure 1 and 2 are not illustrative.**
>
> **A1.** Please refer to the revision of the paper, where we have updated Figure 1 and Figure 2.
>
> **In the new Figure 1**, we highlight the conventional "sparse-to-dense" paradigm involves many handcrafted components (e.g., NMS, score thresholding, label assignment), which inevitably introduce accumulated noise to supervision signals for the student. In contrast, our "dense-to-dense" paradigm abandons intermediate operations and enables more informative and precise supervision for the student.
>
> **In the new Figure 2**, we make the following modifications:
>
> 1. We elaborately describe the NMS procedure of the teacher, including candidate grouping, ranking, and suppressing. It is underlined that dense teacher guidance (i.e., grouping information and candidate rank) is extracted from the teacher's NMS process.
>
> 2. We split the proposed dense-to-dense paradigm into three steps. At first, one-to-one correspondence between samples of teacher and student is identified, getting ready for the following behavior imitation. Then, Inverse NMS Clustering is performed. The grouping information and reserved boxes offered by the teacher's NMS procedure are converted into dense training labels for student samples. Finally, through Rank Matching, rank distributions over clustered candidates are enforced to be consistent between the teacher and student.
>
> **Q2. Table 1 is expected to show more results under different labeling ratios, e.g., 20%, 30%, and 50%.**
>
> **A2.** In the **Partially Labeled Data setting**, sampling 1%, 2%, 5% and 10% images as labeled data is a common practice adopted by most previous works [8, 9, 10, 11, 12, 14, 27, 33, 34]. These works didn't report their performance under other labeling ratios (e.g., 20%, 30%, and 50%). Therefore, to make a comparison under 20%, 30%, and 50% labeling ratios, we implement the previous best method (Soft Teacher) using its source code. We list the table here to show the comparison:
>
> |       methods       |   20%    |   30%    |   50%    |
> | :----------------- | :------ | :------ | :------ |
> | supervised baseline |   32.3   |   34.3   |   36.1   |
> |    Soft Teacher     |   35.2   |   37.3   |   38.1   |
> |   DTG-SSOD (ours)   | **37.1** | **38.7** | **39.4** |
>
> Under all three labeling ratios, our DTG-SSOD consistently surpasses the Soft Teacher by a large margin (i.e., at least **+1.3 mAP**), which validates that our method can generalize well on various labeling ratios.
>
> **Q3. Figure 3 is not illustrative.**
>
> **A3.** In the revision, we have refined Figure 3 and polished the corresponding descriptions (section 4.5). We split the original Figure 3 into two independent figures (i.e., Figure 3 and Figure 4 in the revision) for a better presentation. **The new Figure 3** aims to explain an intrinsic problem in the sparse-to-dense paradigm. Specifically, some pseudo boxes are poorly localized, which will mislead the standard IoU-based label assignment and cause noisy training labels for the student. In contrast, without the dependency on pseudo boxes, our method can effectively alleviate this problem. We show two examples **in the new Figure 3**, where the *identical* student proposals will obtain *different* training labels in the two paradigms.  Compared with the sparse-to-dense paradigm,  our dense-to-dense paradigm obviously offers more precise training labels for the student. In the sub-fig (1), the sparse-to-dense paradigm treats the **poor** proposal (in white) as a positive sample. The proposal has a relatively high IoU value (i.e., 0.6) with the coarse pseudo box (in red),  reaching the requirement of positive samples in the standard IoU-based label assignment. Compared with it, our Inverse NMS Clustering takes teacher predictions into consideration and succeeds in suppressing this misleading positive. Concretely, with the low confidence (i.e., 0.3) predicted by the teacher, this proposal will be a clear background sample in the teacher's NMS. Another example is shown in the sub-fig (2). Due to the poor localization of the pseudo box, a relatively precise proposal is wrongly assigned as a negative sample by the sparse-to-dense paradigm. However, our method can avoid this problem by taking the NMS behavior of proposals into consideration.
>
> On the other hand, we also exhibit an example **in the new Figure 4**, to demonstrate that the teacher is better at modeling candidate rank than the student. Specifically, between the two candidate boxes shown in the figure, the teacher ranks the one with more precise localization first, while the student fails. Candidate rank predicted by the teacher can serve as beneficial dense guidance, which is missing in the conventional sparse-to-dense paradigm.

---

### Meta-Review · Area_Chair_YLpo · 2022-09-06

**Recommendation:** Accept
**Confidence:** Certain

**Metareview:**


 This paper proposes a dense-to-dense semi-supervised object detection method, where the teacher's NMS is used to guide the clustering and ranking of bounding box candidates from the student. This is motivated from potential noise resulting from sparse-to-dense pseudo-label supervision in existing methods. Results are shown on standard semi-supervised object detection benchmarks, with improvements over the current state of art.

The reviewers all thought that the paper had an interesting idea, strong results, and thorough experiments, ablations, and analysis. Some concerns included generalization to other architectures (e.g. DETR or single-stage CNN), comparison to feature distillation, and poor communication especially through the figures. The rebuttal provided answers to these, including new experiments showing generalization to a single-stage method, and all reviewers have recommended acceptance (and the reviewer with borderline accept mentioned it is a good paper). As a result, I recommend accepting this paper as it provides an interesting new contribution to the common mean teacher paradigm. I highly encourage the authors to add new elements that came out in the rebuttal, especially generalization to single-stage methods and failure cases.

**Award:**

No

---

### Decision · Program_Chairs · 2022-09-14

Accept